# Polyparasitic Infections: Associated Factors and Effect on the Haemoglobin Level of Children Living in Lambaréné Remote and Surrounding Rural Areas from Gabon—A Cross-Sectional Study

**DOI:** 10.3390/tropicalmed10080218

**Published:** 2025-07-31

**Authors:** Paul Alvyn Nguema-Moure, Bayode Romeo Adegbite, Moustapha Nzamba Maloum, Jean-Claude Dejon-Agobé, Roméo-Aimé Laclong Lontchi, Yabo Josiane Honkpehedji, Danny-Carrel Manfoumbi Mabicka, Christian Chassem-Lapue, Pavel Warry Sole, Stephane Ogoula, Fabrice Beral M’Baidigium, Jenny Mouloungui-Mavoungou, Michael Ramharter, Peter Gottfried Kremsner, Ayôla Akim Adegnika

**Affiliations:** 1Centre de Recherches Médicales de Lambaréné (CERMEL), Lambaréné BP 242, Gabonogoulastephane09@gmail.com (S.O.); aadegnika@cermel.org (A.A.A.); 2Institute of Tropical Medicine, Universitätsklinikum Tübingen, Universität Tübingen, 72074 Tübingen, Germany; 3Leiden University Center for Infectious Diseases (LUCID), Leiden University Medical Center, 2333 ZA Leiden, The Netherlands; 4Bernhard Nocht Institute for Tropical Medicine, 20359 Hamburg, Germany; 5The German Center for Infection Research (DZIF), Tübingen Site, 72076 Tübingen, Germany

**Keywords:** haemoglobin, neglected tropical diseases, polyparasitic infections, soil-transmitted helminths infections, urinary schistosomiasis

## Abstract

Background: Polyparasitic infections remain widespread in endemic regions, yet its contributing factors and health impact are not well understood. This study aims to estimate the prevalence and associated factors and examines the effect of polyparasitic infection on haemoglobin levels among children. Methods: A cross-sectional study was conducted in Lambaréné, Gabon, among children aged 2–17 years from November 2019 to December 2020. Haemoglobin levels, environmental conditions, and sociodemographic data were collected. Stool, urine, and blood samples were analysed using light microscopy for parasite detection. Factors associated with polyparasitism were explored. Results: Out of 656 participants, 65.4% had at least one infection, with intestinal protozoa species (21.3%), *Trichuris trichiura* (33%), *Ascaris lumbricoides* (22%), *Schistosoma haematobium* (20%), and *Plasmodium falciparum* (10%) being the most common. Polyparasitic infection was identified in 26% of children, mostly as bi-infections (69.2%), and was negatively associated with haemoglobin levels (β = −0.06). Conclusions: These findings emphasise the burden of polyparasitic infections and adverse health effects in Lambaréné, Gabon.

## 1. Introduction

Polyparasitic infections refer to the presence of two or more parasitic species within the same host [1]. This is common in many endemic areas across tropical and subtropical regions [2,3]. Malaria, urogenital schistosomiasis, and soil-transmitted helminth (STH) infections are the most prevalent and endemic parasitic diseases in these regions. In children, these infections can lead to severe anaemia, affect growth and cognitive development, and may even result in death [4,5,6]. Protozoa like *Blastocystis hominis*, *Giardia duodenalis*, among others [7,8], also contribute to polyparasitic infections. In 2023, approximately 253.8 million people, 85% of whom reside in sub-Saharan Africa, were infected with *Schistosoma* spp., with 124.4 million of them being school-age children [5]. Over 1.5 billion people are infected with STH species, including 568 million school-age children living in high-transmission areas [5]. Global trends for malaria in 2024 reported approximately 263 million cases worldwide, with around 597,000 deaths, 90% of which occurred in the African region [9]. These infections often coexist (polyparasitic infections), leading to increased morbidity, including immune hyperresponsiveness, increased susceptibility to other infections, poor immunogenicity of vaccines [10], and adverse pregnancy outcomes [10]. Polyparasitic infection is also associated with malnutrition, impaired cognitive development, splenomegaly, anaemia, and leucocytosis in school-age children [11]. This underscores the importance of controlling parasitic diseases, particularly in endemic areas.

The Lambaréné region in Gabon is known to be endemic to many parasitic infections, such as malaria, schistosomiasis, and intestinal soil-transmitted helminths [12]. This region is therefore eligible for mass drug administration (MDA) using praziquantel and a benzimidazole to treat school-aged children. The area is also known to be endemic for filarial parasites [13,14]. Considering that higher *Loa loa* microfilariae parasitaemia causes an increased risk of encephalopathy following albendazole treatment, there is hesitancy about the use of albendazole in mass drug administration (MDA). In such settings, understanding the epidemiology of parasitic co-infection is crucial for determining the frequency of polyparasitic infection and for tailoring appropriate control strategies. Identifying factors associated with polyparasitic infection will help pinpoint individuals at risk and apply triage strategies, such as test and treat, before any albendazole MDA. Additionally, studies in pregnant women have reported pregnancy outcomes including low birth weights as adverse effects of polyparasitic infections [15]. To our knowledge, the adverse effects (such as anaemia) of polyparasitic infection in school-age children have not yet been reported in the Lambaréné region. This study aims to estimate the prevalence and associated factors for polyparasitic infection and its effect on haemoglobin levels in school-aged children in Lambaréné and the surrounding villages.

## 2. Materials and Methods

### 2.1. Study Design

This study was a cross-sectional sub-analysis of data collected from November 2019 and December 2020. The original data came from a clinical trial aiming to assess the efficacy of a new regimen for the treatment of soil-transmitted helminths in school-age children, aged 2 to 17 years, from Lambaréné (both semi-urban and surrounding rural areas in Moyen-Ogoouée province) and remote areas of Tsamba-Magotsi (Ngounié province) [16]. The purpose of this sub-analysis was to detail the epidemiological profile of parasitic infections within this population.

### 2.2. Procedures

The selection of participants was conducted through the engagement of the local leader. The study aim was presented to the guardians of participants expressing willingness to participate. Volunteers provided written informed consent (obtained from parents or legal guardians) before their inclusion in the study. Upon inclusion, participants were given two separate labelled containers and instructed to provide field workers with urine and stool samples the following day. Additionally, blood samples were collected at home from participants who had previously provided stool and/or urine samples. The samples were primarily used to assess intestinal parasite infections (such as STH and intestinal protozoa infections) and urogenital schistosomiasis. Blood was also collected to assess the endemicity of blood parasite infections, such as plasmodium infection and filariasis. The information on demographic, socioeconomic, household, and behavioural characteristics was collected from participants and parents/guardians using a questionnaire.

### 2.3. Study Population

Participants were invited to participate. Stool and urine samples were collected from children whose parents provided informed consent and who were included in the study. The participants included in the analysis had no history of anthelmintic treatment (in the past 2 months) or antimalarial treatment (in the past 2 weeks). We excluded participants who did not provide any samples.

### 2.4. Ethical Consideration

The study protocol was approved by the Institutional Ethics Committee from CERMEL (CEI 007/2019) and the National Ethics Committee of Gabon (PROT N°0084/2019/PR/SG/CNER).

Parents or legal representatives of participants gave a written consent form, and the children up to 13 years old gave their assent forms before any study procedure. The study was conducted in line with the Good Clinical Practice (GCP) principles of the International Conference on Harmonisation (ICH) and the Declaration of Helsinki.

### 2.5. Study Site and Areas

The study was carried out in Lambaréné at the Centre de Recherches Médicales de Lambaréné. The study site included Lambaréné city (a semi-urban area), rural areas surrounding Lambaréné (villages located north and south of Lambaréné in the Ogooué et Lacs department, in the Gabon country), and remote areas of Tsamba-Magotsi in the Ngounié province of Gabon. These areas are known to be endemic for parasitic infections such as malaria, STH, and schistosomiasis. Lambaréné city is located 60 km south of the equator and is surrounded by several villages along the N1 national road, both to the north and south of the city. Lambaréné has approximately 45,000 inhabitants, while the surrounding rural areas have about 3500 inhabitants, according to a national survey in 2016 [17]. In Tsamba-Magotsi (with 14,875 inhabitants), the study was conducted in rural areas along the N1 national road, from the Tchad villages to Oyenano village, including Koinia village located on the Sinadara road, which constitutes remote rural areas relative to Lambaréné. These areas are close to Fougamou City, the second-largest town of the Ngounié province (Figure 1). Previous research has documented the presence of infectious diseases such as urinary schistosomiasis and soil-transmitted helminth (STH) infections in these areas [17].

### 2.6. Sample Size Calculation

The sample size was determined using the StatCalc software from Epi Info 7.2.6.0. The parameters used for the calculation included a confidence level of 95%, a power of 80%, an estimated prevalence of polyparasitic infection of 7.8% in Lambaréné [17], and a margin of error of 5%. Based on these parameters, the minimum sample size was 550. This sample size was increased to 700 to consider missing data, cases of consent withdrawal, or refusal to provide blood, stool, and urine samples.

### 2.7. Participant Recruitment and Sample Collection

From November 2019 to August 2020, all volunteers were recruited for the study in each study area. The fieldworkers visited participants’ households to invite them to join the study. After obtaining informed consent, sociodemographic information and social habits were collected. We provided eligible participants with plastic containers and asked them to provide a stool sample and a urine sample (10 mL minimum). In addition, 1 mL of blood was collected and used for the diagnosis of blood filariasis and plasmodium infection using microscopy. Samples were collected from participants’ homes and tested at the research centre, Centre de Recherches Médicales de Lambaréné, within 2 h following sample reception at the laboratories.

### 2.8. Laboratory Tests

#### 2.8.1. Direct Examination from Stool Sample

This technique was applied to identify the presence of intestinal protozoa species (IPS) in the stool. After homogenization of the sample with a spatula, direct examination using around 2 mg of the filtered stool sample was spread (using a saline solution from 0.9% sodium chloride) on a slide microscope, where a drop of 2% Lugol’s iodine was added to colour the specimen and facilitate the identification under a light microscope [18]. Each sample slide was read until 50 microscope fields (40×) minimum before assessing the positivity or negativity of the preparation.

#### 2.8.2. Kato–Katz Technique

A double Kato–Katz (two-slide preparation was performed) technique was used for the detection and quantification of STH eggs in stool samples, as we described previously [19]. Briefly, a total of 41.7 mg of stool was used from each stool sample. The smears were performed on duplicate slides to improve the sensitivity of the test to find eggs of *A. lumbricoides*, *T. trichiura*, and hookworms. Slides were examined by light microscopy after 10 min of preparation.

#### 2.8.3. Coproculture Technique

To detect hookworms and *Strongyloides stercoralis* larvae in stool samples, the coproculture technique was applied. Approximately 1 g of the stool sample was spread on a slide and covered with filter paper. This device was placed in a Petri dish, and approximately 20 mL of Normal saline solution: 0.9% NaCl (or sterile tap water) was added to the bottom of the dish. The preparation was then incubated for 7 days at a temperature of 22–28 °C. After incubation, approximately 10 mL of water was collected with a syringe and filtered using a Whatman nucleopore membrane (12 µm pore size) and a Swinnex (Merck, Darmstadt, Germany) filter holder. The filter paper was then examined under a microscope with a drop of 2% Lugol’s iodine solution. The sample was considered positive if at least one larva was detected (Figure 2) [20].

#### 2.8.4. Harada–Mori Culture Technique

In this technique, we used a filter paper to isolate the also infective third-stage larvae. One gram of fresh faeces was placed at the centre of a sterile filter paper strip, which was then inserted into a 15 mL centrifuge tube containing 4 mL of distilled water. The tapered end of the strip touched the bottom of the tube, while the faecal spot remained above the water level to prevent direct immersion. The tube was sealed with a screw cap and incubated upright at 25–28 °C for 10 days. After incubation, the fluid at the bottom of the tube was centrifuged, and a smear was prepared from the resulting pellet. Microscopic examination was performed using a 10× objective lens to identify motile third-stage larvae [21]. To differentiate between the *Strongyloides stercoralis* and hookworm larvae in both culture techniques (coproculture and Harada–Mori), we applied the following morphological criteria. Hookworm L3 larvae were identified by their pointed tail, short oesophagus, and overall smooth, tapered posterior end. *S. stercoralis* L3 larvae were distinguished by a notched tail, a long oesophagus with a 1:1 ratio to the intestine, and more vigorous motility [20].

#### 2.8.5. Urine Filtration Technique

Urine samples collected were analysed using the urine filtration technique. Briefly, 10 mL for each urine sample, urine filtration (using Swinnex^®^ kit and nucleopore Whatman^®^ membrane) was performed for the detection of *S. haematobium* eggs (using a 12 µm pore size Whatman^®^ membrane) as described elsewhere [22]. After examination of the nucleopore Whatman^®^ membrane under the microscope, the mean number of eggs was expressed in 10 mL of filtered urine.

#### 2.8.6. Blood Sample Examination

Blood samples were collected by venepuncture in EDTA tubes (2.7 mL). Before microscopic examination, haematological parameters were analysed using the Yumizen H500 automated system (Horiba Medical, Montpelier, France). Haematological printing results were sent within 24 h to the clinician for assessment and decision.

Additionally, saponin lysis (2%) and the Lambaréné technique were applied to the collected blood samples. The saponin lysis method was used for the identification and quantification of blood filarial parasites, as previously described [23].

Malaria diagnosis was conducted using conventional thick blood smear (Giemsa-stained at 10%) microscopy [22].

#### 2.8.7. Quality Control from Microscopy Results

Two microscopists examined all the slides prepared for internal quality control. The two microscopy readings should agree at ±80%. In the case of discordant readings, a third reading was required, and the final results considered the readings that had an agreement of ±80%. Following the standardised operational procedures established by the laboratory institution, the parasite count was calculated by averaging the combined results from both readers.


Case definition


We considered any participant to be infected if their slide preparation sample presented at least one form of parasite.We considered a polyparasitic infection case to be any participant infected by at least two parasite groups of parasite species (the groups of parasite species considered were STH, IPS, *Schistosoma* spp., *Plasmodium* spp., or microfilariae worm species).

### 2.9. Statistical Analysis

All data were recorded in the Red Cap database, exported in Excel (2016 version), and analysed using StataIC 16. Percentages and frequencies were used to express categorical variables with a confidence interval (CI). The normality of continuous variables was assessed using the Kolmogorov–Smirnov and the Shapiro–Wilk tests. The median and interquartile range (IQR) were used for continuous variables. The logistic regression was used to identify associated factors with polyparasitic infection. Additionally, a linear regression was performed to examine the effect of multiple variables on haemoglobin levels. Confidence intervals were set at 95% ([95% CI]), and *p*-values were considered significant at values less than or equal to 0.05 (5%).

## 3. Results

### 3.1. Characteristics of the Study Population

Out of the 700 participants invited to take part in the study, 656 were enrolled (Figure 3). Both sexes were recruited in a similar proportion (*n* = 328, 50.0%). The median age was 7 years (IQR = 4–11). The median haemoglobin value was 10.4 g/dL (IQR = 9.4–10.3). Most (*n* = 369, 56.3%) of the participants lived in Lambaréné and the surrounding villages (Table 1).

### 3.2. The Prevalence of Species Mono-Infection and Polyparasitic Infection

A total of 429 children (65.4%) had at least one parasitic infection, and 169 children (25.8%, 95% CI: 21.9–28.4) experienced polyparasitic infection. Most polyparasitic infection cases involved bi-infections (69.2%, 95% CI: 61.9–75.7, *n* = 117), as shown in Figure 3.

Regarding parasitic infections by species, the prevalence was as follows:

*Trichuris trichiura*: 32.5% (213/656), 95% CI: 29.0–36.1, *Ascaris lumbricoïdes*: 22% (144/656), 95% CI 19.0–25.3), *Schistosoma haematobium*: 20.0% (131/656), 95% CI: 1708–23.2; *Plasmodium falciparum*: 10.0%, 95% CI: 7.6–12.2. The prevalence of infections by *Loa loa* and *Mansonella perstans* (*n* = 656) was 3.0% 95% CI: 1.4–3.8, and 5.0%, 95% CI: 3.5–6.8, respectively. The prevalence of intestinal protozoa was 21.34% (95% CI: 18.2–24.4).

*Blastocystis hominis* (5.5%, 95% CI: 4.0–7.5, *n* = 36) and *Giardia duodenalis* (4.4%, 95% CI: 3.0–6.2, *n* = 29) were frequently found in the analysed samples, while a notable prevalence of other species like *Entamoeba coli* (3.8%, 95% CI: 2.4–5.3, *n* = 25), *Endolimax nana* (2.7%, 95% CI: 1.7–4.3, *n* = 18), *Entamoeba histolytica/dispar* (2.4%, *n* = 16), and *Iodamoeba bütschlii* (2.4%, *n* = 16) were also detected (Table 2).

### 3.3. Factors Associated with Polyparasitic Infection and Its Effect on Haemoglobin Level

Individuals from the surrounding rural areas (Ogooué et Lacs) exhibited a significantly lower odds ratio (OR 0.16, *p* < 0.001 in bivariable analysis; OR 0.32, *p* = 0.01 in multivariable analysis), indicating a reduced likelihood of polyparasitic infection compared to urban and remote areas. Conversely, river water consumption was strongly associated with polyparasitic infection (OR 4.45, *p* < 0.001 in bivariable analysis; OR 1.98, *p* = 0.04 in multivariable analysis), suggesting an increased risk. Additionally, non-conventional sanitation methods showed a significant association with polyparasitic infection (OR 2.64, *p* < 0.001 in bivariable analysis; OR 2.06, *p* = 0.003 in multivariable analysis), emphasising the impact of inadequate sanitation on infection rates (Table 3).

Individuals with polyparasitic infection had significantly lower haemoglobin levels (β = −0.48, *p* = 0.03 in bivariable analysis; β = −0.60, *p* = 0.01 in multivariable analysis). Living in a wooden house was associated with lower haemoglobin levels (β = −4.1, *p* = 0.03 in multivariable analysis). Factors such as sex, age, geographic location, washing hands before meals, water source, barefoot walking, and sanitation did not show strong associations with haemoglobin levels, as their regression coefficients did not reach statistical significance (Table 4).

## 4. Discussion

This study investigated parasitic infections among school-aged children in Lambaréné and its surrounding areas. The results showed that 64.4% of the studied children were infected with at least one parasite. This high prevalence of parasitic infection is consistent with findings from multiple studies conducted in tropical and subtropical regions [24,25] where parasites are endemic and socioeconomic conditions facilitate their transmission. The prevalence of polyparasitic infection was 26% (95% CI: 23.0–29.2), highlighting the occurrence of co-infection. This finding is consistent with other regional studies that have reported high rates of polyparasitic infection among school-aged children [26,27]. This prevalence is relatively higher than the 19.9% prevalence of polyparasitic infection reported in Cameroon in 2021, among children living in rural areas (*n* = 638 participants) [28]. This difference may be due to environmental conditions and the relatively higher parasite burden in our study.

Most cases of polyparasitic infection were dual infections (69.2%), which is consistent with other studies indicating that mixed infections often involve two species [29]. The prevalence of *T. trichiura* (33%), *A. lumbricoides* (22%), and *S. haematobium* (20.0%) was particularly high, reflecting the high endemicity of these parasites in the study areas, where sanitary and environmental conditions favour their spread; *T. trichiura*, *A. lumbricoides*, and *S. haematobium* are endemic in Lambaréné and the vicinity areas. Other parasites, such as *P. falciparum*, *Loa loa*, and *M. perstans*, also showed significant prevalence rates, although lower than those of the first three species. Additionally, the presence of intestinal pathogenic protozoa species, such as *B. hominis* (14.4%) and *G. duodenalis* (11.6%), in stool samples was notable. These data confirm the endemicity of soil-transmitted helminths and urogenital schistosomiasis in the study area [12,17,29]. They also highlight the significant prevalence of intestinal pathogenic protozoan species like *B. hominis* and *G. duodenalis* in the study area, including among asymptomatic individuals, as previously reported in other localities (urban and rural) in Gabon [30,31,32], suggesting the need for increased efforts to control and eliminate these infections.

The bivariable and multivariable analyses highlight several factors associated with polyparasitic infection, a condition involving the simultaneous presence of multiple parasitic infections. First, we observed a strong association between polyparasitic infection and geographical area. Individuals from remote rural areas (Tsamba-Magotsi and Ogooué et Lacs) were more likely to have polyparasitic infection compared to those from rural Ogoouée et Lacs. The multivariable analysis showed that participants from Tsamba-Magotsi were more exposed to polyparasitic infection (aOR = 1.1, 95% CI: 0.4–2.8, *p* < 0.8). The difference in polyparasitic infection rates can be explained by access to health infrastructure, as highlighted by Branda F. and collaborators in the case of neglected tropical diseases [33]. People living far from Lambaréné city do not benefit as much as those in urban areas, where hospitals are located several kilometres away, which may limit access to treatment and the prevention of parasitic infections.

Additionally, access to clean water was a significant determinant of polyparasitic infection. Participants using river water were found to be at high risk of polyparasitic infection (OR = 4.4, 95% CI: 2.5–8.0, *p* < 0.001). This result aligns with studies from other towns in Gabon in 2014 [31] and more recently in 2023 [33]. Similar findings were observed in the Likoko area in Cameroon [28], emphasising the importance of improving water sanitation in surrounding and remote rural areas.

The association between non-conventional toilets (e.g., open-pit latrines) and polyparasitic infection was significant in both analyses. The ORs for non-conventional toilets were 2.64 [95% CI: 1.8–3.7], *p* < 0.001 in bivariable analysis and 2.06 (95% CI: 1.2–3.3, *p* = 0.003) in multivariable analysis, indicating that improper sanitation systems were associated with increased polyparasitic infection. This finding is consistent with other studies highlighting the role of inadequate sanitation in the transmission of parasitic diseases [7,32,33].

The effect of polyparasitic infection on haemoglobin levels, as assessed by multiple linear regression, revealed a significant negative relationship. Participants living in Planck houses (non-durable housing structures) exhibited a significantly greater reduction in haemoglobin levels (β = −4.1, 95% CI: −7.9 to −0.34, *p* = 0.03). This could be related to poorer living conditions, including increased exposure to environmental risk factors such as poor sanitation and vector-borne diseases, which exacerbate the impacts of parasitic infections on health [34,35]. Additionally, a statistically significant reduction in haemoglobin levels was observed in participants with polyparasitic infection (β = −0.60, 95% CI: −1.1 to −0.12, *p* = 0.01). This aligns with studies such as those by Bustinduy et al. (2013) [34] and the World Health Organisation (WHO) report [5] which demonstrated that multiple parasitic infections are associated with anaemia, potentially due to chronic inflammation or blood loss from helminths, which can impair red blood cell production.

The strengths of the study lie in its rigorous and detailed approach to investigating parasitic infections and polyparasitic infections, as well as its focus on relevant socioeconomic and environmental factors. This study highlights the health consequences of these infections, emphasising the importance of improving sanitation and access to clean water in rural areas to reduce the burden of parasitic diseases. These findings have significant public health implications and advocate for the implementation of effective health policies to combat parasitic infection. The main limitation of our study is a cross-sectional design, meaning data were collected at a single point in time. Consequently, it is not possible to establish causal relationships between the factors studied (such as housing type, water source, or barefoot walking) and the occurrence of parasitic infections. Additionally, habits related to barefoot walking and handwashing frequency were self-reported, which may introduce recall bias—participants may not accurately remember their habits or may underreport or overestimate them. The study used haemoglobin levels to assess anaemia but did not consider other nutritional biomarkers that could be relevant for evaluating the impact of parasitic infections on children’s health. Finally, the study participants were not randomly selected, which may lead to selection bias.

## 5. Conclusions

This study emphasises the extent of polyparasitic infection among school-aged children in Lambaréné city and surrounding rural areas in the Ogooué-et-Lacs department and remote regions in the Tsamba-Magotsi department. It highlights the significant public health burden of polyparasitic infection, especially in rural settings with inadequate sanitation and water resources. The results advocate for integrated strategies targeting environmental, behavioural, and healthcare aspects to address parasitic infections, improve haemoglobin levels, and reduce associated morbidity in the affected populations.

## Figures and Tables

**Figure 1 tropicalmed-10-00218-f001:**
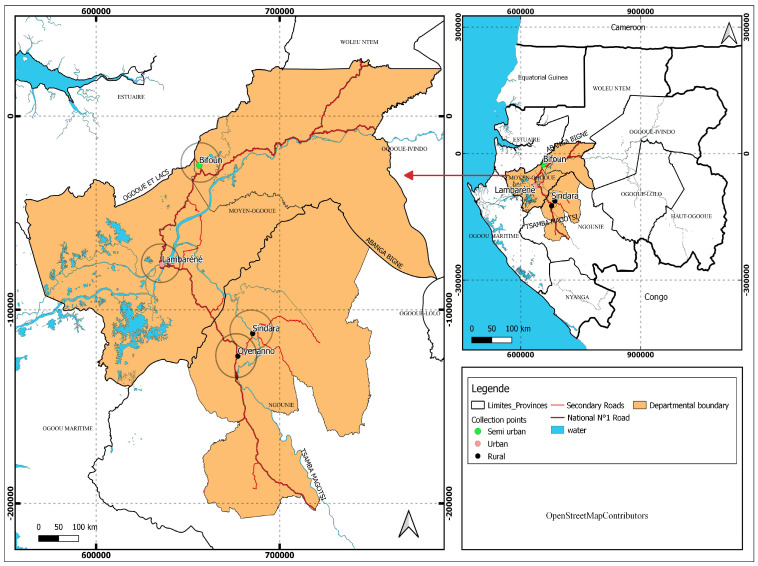
Study map showing Lambaréné, the rural areas surrounding (Ogooué et Lacs department), and the remote areas around Lambaréné (Tsamba-Magotsi department) where the samples had been collected.

**Figure 2 tropicalmed-10-00218-f002:**
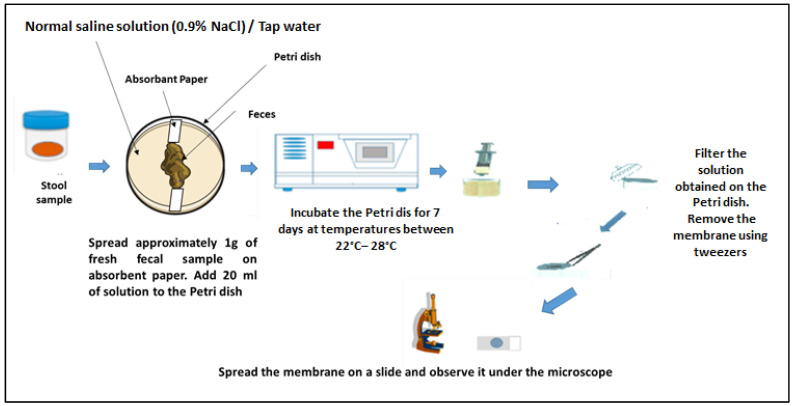
Coproculture technique. Adapted from an open-source web image.

**Figure 3 tropicalmed-10-00218-f003:**
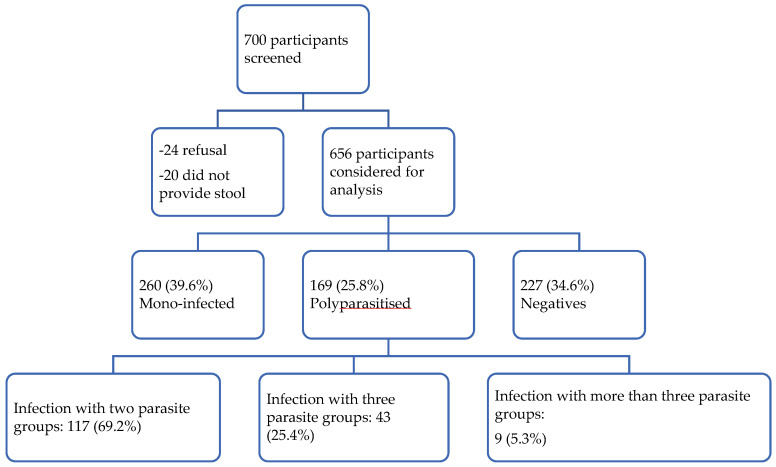
Study flow chart.

**Table 1 tropicalmed-10-00218-t001:** General characteristics of the 656 participants of the study population.

Characteristic	Frequency (%)	[95% CI]
**Age group (years)**		
2 to 5	238 (36.3%)	[32.7–40.3]
6 to 10	249 (38.0%)	[34.3–41.7]
11 to 17	169 (25.8%)	[25.8–29.2]
**Areas**		
Lambaréné (semi-urban)	59 (9.0%)	[7.0–11.5]
Sur. areas LBN (Ogooué et Lacs)	369 (56.3%)	[52.4–60.0]
Remote areas LBN (Tsamba-Magotsi)	227 (34.6%)	[31.0–38.3]

Sur. areas to LBN: rural areas surrounding Lambaréné in the Ogooué et Lacs department; Remote areas to LBN: rural areas remote to Lambaréné located in the Tsamba-Magotsi department.

**Table 2 tropicalmed-10-00218-t002:** Prevalence and parasite count means of soil-transmitted helminths, *Schistosoma haematobium*, *P. falciparum*, and blood filarials in the study population.

Parasite Species (*n* = 656)	Prevalence (%)	[IC 95%]	Parasite Count
Any STH	280 (42.7)	[39.0–46.4]	-
*Trichiurus trichiurus*	213 (32.5)	[29.0–36.1]	1513 EPG
*Ascaris lumbricoides*	144 (21.9)	[18.9–25.3]	43825 EPG
*Strongyloides stercoralis*	64 (9.8)	[7.5–12.4]	12 Larvae
Hookworms	62 (9.5)	[7.4–11.9]	372 EPG
*Schistosoma haematobium*	131 (19.9)	[34.7–45.3]	136 eggs/10 mL
Any blood filaria species	49 (7.5)	[7.5–9.40]	-
*Mansonella perstans*	32 (4.9)	[3.2–6.8]	5 P/mL
*Loa loa*	17 (2.6)	[1.4–3.8]	6 P/mL
Blood protozoa (*P. falciparum)*	65 (9.9)	[7.6–12.2]	1582 P/µL
Any intestinal protozoa species	140 (21.3%)	[18.2–24.4]	-
*Blastocystis hominis*	36 (5.5%)	[4.0–7.5]	-
*Giardia duodenalis*	29 (4.4%)	[3.0–6.2]	-
*Entamoeba coli*	25 (3.8%)	[2.4–5.3]	-
*Iodamoeba bütschlii*	16 (2.4%)	[1.5–3.9]	-
*Endolimax nana*	18 (2.7%)	[1.7–4.3]	-
*Entamoeba histolytica/dispar/hartmanni*	16 (2.4%)	[1.5–3.9]	-

EPG: eggs per gram; P/µL: Parasites per microlitre; -: not applicable; P/mL: parasites per millilitre; eggs/10 mL: egg count per 10 millilitres.

**Table 3 tropicalmed-10-00218-t003:** Bivariable and multivariable analysis to determine the associated factors with polyparasitic infection.

	Polyparasitic Infection		Bivariable Analysis			Multivariable Analysis		
Characteristic	No *n* = 487	Yes *n* = 169	OR	95% CI	*p*-Value	OR	95% CI	*p*-Value
**Age (years)**	7.0 (4.0, 10.0) ^1^	8.0 (5.0, 11.0) ^1^	1.02	0.97, 1.06	0.45	1.05	0.93, 1.19	0.4
**Sex**								
F	236 (73%)	89 (27%)	—	—		—	—	
M	251 (76%)	80 (24%)	0.85	0.59, 1.20	0.35	0.85	0.58, 1.26	0.4
**Areas**								
Lambaréné	30 (51%)	29 (49%)	—	—		—	—	
Remote areas LBN (Tsamba-Magotsi)	132 (58%)	96 (42%)	0.75	0.42, 1.34	0.33	0.75	0.29, 1.98	0.6
Sur. North and Sud LBN (Ogooué et Lacs)	325 (88%)	44 (12%)	0.14	0.08, 0.25	<0.001	0.22	0.09, 0.55	0.001 *
**Washing hands before meals**								
Often	26 (70%)	11 (30%)	—	—		—	—	
Rarely	76 (78%)	21 (22%)	0.65	0.28, 1.57	0.33	1.06	0.34, 3.36	>0.9
Sometimes	385 (74%)	137 (26%)	0.84	0.41, 1.82	0.64	0.89	0.35, 2.31	0.8
**Water source**								
Tap water	98 (84%)	19 (16%)	—	—		—	—	
Mixed	285 (84%)	56 (16%)	1.01	0.58, 1.83	0.96	0.83	0.43, 1.68	0.6
River	104 (53%)	94 (47%)	4.66	2.70, 8.39	<0.001	2.09	1.06, 4.25	0.038 *
**House type**								
Cement brick	14 (47%)	16 (53%)	—	—		—	—	
Planck house	473 (76%)	153 (24%)	0.28	0.13, 0.59	<0.001	0.59	0.20, 1.73	0.3
**Barefoot walking**								
Often	218 (74%)	77 (26%)	—	—		—	—	
Rarely	58 (79%)	15 (21%)	0.73	0.38, 1.34	0.33	0.57	0.14, 2.29	0.4
Sometimes	211 (73%)	77 (27%)	1.03	0.71, 1.49	0.86	1.65	0.77, 3.53	0.2
Sanitation								
WC or Latrine	332 (81%)	77 (19%)	—	—		—	—	
Non-conventional	155 (63%)	92 (37%)	2.56	1.79, 3.67	<0.001	2.00	1.25, 3.22	0.004 *

^1^ Median (Q1, Q3); OR: Odds Ratio, CI: Confidence Interval; LBN: Lambaréné city; Sur. North and Sud LBN: rural areas surrounding Lambaréné in the Ogooué et Lacs department; Remote areas to LBN: rural areas remote to Lambaréné located in the Tsamba-Magotsi department; WC; water closet; * *p*-value: significant.

**Table 4 tropicalmed-10-00218-t004:** Bivariable and multiple linear regression to assess the effect of polyparasitic infection on the haemoglobin level among the polyparasitised study participants.

	Bivariable Analysis			Multivariable Analysis		
Characteristic	β	[95% CI]	*p*-Value	β	[95% CI]	*p*-Value
**Polyparasitic infection**						
No	Ref	—		—	—	
Yes	−0.48	[−0.92–−0.04]	0.03 *	−0.60	[−1.1–−0.12]	0.01 *
**Sex**						
F	Ref	—		—	—	
M	−0.21	[−0.65–0.24]	0.36	−0.40	[−0.86–0.07]	0.09
**Age (years)**	−0.04	[−0.11–0.02]	0.16	−0.01	[−0.18–0.16]	0.92
**Areas**						
Lambaréné	Ref	—		—	—	
Remote areas LBN (Tsamba-Magotsi)	0.62	[−1.3–2.5]	0.52	2.4	[−0.27–5.1]	0.07
Sur. areas LBN (Ogooué et Lacs)	0.21	[−1.7–2.1]	0.83	1.9	[−0.86–4.6]	0.18
**Washing hands before meals**						
Often	Ref	—		—	—	
Rarely	1.3	[−0.41–2.9]	0.14	1.1	[−1.1–3.3]	0.32
Sometimes	0.93	[−0.64–2.5]	0.24	0.94	[−0.91–2.8]	0.32
**Water source**						
Tap water	Ref	—		—	—	
Mixed	0.73	[−0.10–1.6]	0.08	0.60	[−0.32–1.5]	0.20
River	0.39	[−0.45–1.2]	0.36	0.22	[−0.84–1.3]	0.68
**House type**						
Cement brick	Ref	—		—	—	
Planck house	−1.1	[−3.8–1.6]	0.43	−4.1	[−7.9–−0.34]	0.03 *
**Barefoot walking**						
Often	Ref	—		—	—	
Rarely	−0.29	[−1.1–0.46]	0.44	0.21	[−1.4–1.9]	0.80
Sometimes	−0.39	[−0.87–0.08]	0.11	0.02	[−0.85–0.89]	0.97
**Sanitation**						
WC or Latrine	Ref	—		—	—	
Non-conventional	0.23	[−0.22–0.68]	0.32	0.12	[−0.40–0.65]	0.64

CI: Confidence interval, β: regression coefficient; Ref: reference; WC: water closet; *: Significant *p*-value; LBN: Lambaréné city; Sur. Areas to LBN: rural areas surrounding Lambaréné in the Ogooué et Lacs department; Remote areas to LBN: rural areas remote to Lambaréné located in the Tsamba-Magotsi department.

## Data Availability

All relevant data are within the paper. The datasets used and/or analysed during the current study are available from the corresponding author on reasonable request.

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
