# Peer review of "Polyparasitic Infections: Associated Factors and Effect on the Haemoglobin Level of Children Living in Lambaréné Remote and Surrounding Rural Areas from Gabon—A Cross-Sectional Study"

_tropicalmed, 2025, doi:10.3390/tropicalmed10080218_

Round 1

Reviewer 1 Report

Comments and Suggestions for Authors

Parasitosis can be understood as an indirect index of poor socio-economic development. Intestinal parasitic infections pose a significant public health risk, particularly among vulnerable groups such as children. Endoparasites in children are generally associated with challenges in child development (height, weight, and cognitive development). Addressing these issues is a major challenge in the context of One Health. The present paper was designed to estimate the prevalence and associated factors for polyparasitism and its effect on hemoglobin levels in school-aged children in Lambaréné and the sur- rounding villages in Gabon, Central Africa. Several publications on children in Gabon have already demonstrated high rates of polyparasitism in the analyzed population, including double or triple infections. Other data has also revealed a link between polyparasitism and anemia. Several studies aimed to determine the prevalence of co-infection with malaria and intestinal parasites and assess its association with anemia in school-aged children from rural and urban settlements in Gabon. To summarize, the issue raised by the authors is not unique to Gabon. For instance, the results of an epidemiological survey conducted in Gabonese provinces were recently published (see the following link: https://doi.org/10.1371/journal.pntd.0013161). In addition to the risk factors cited for the prevalence of parasitic diseases among schoolchildren, guardians or parents' level of education is also a relevant factor. Was this question included in the questionnaire?

The title accurately and keywords reflect the topic of the paper, which is clearly organized in several sections, including abstract, introduction, descriptions of methods, results with tables of data, discussion, conclusion and references. The manuscript is well-written and easy to understand. Regarding references, citations is partially adequate because recent study was not indicated https://doi.org/10.1371/journal.pntd.0013161).

The manuscript raised some questions.

Line 77 – of soil-transmitted helminth instead of sol soil-transmitted helminth

Line 92 – malaria and filariasis

Line 148- malaria?

Line 152- Direct examination of stool samples is a technique with low sensitivity. However, it can be useful for detecting trophozoites in diarrhoeal samples of intestinal protozoa, as well as motile Strongyloides stercoralis larvae.

Lines 167-179 - How many coproculture or Harada-Mori techniques were performed? 676?

Lines 187-189 – The information should be detailed if anyone wants to reproduce the techniques described here. However, the authors do not specify the criterion used to differentiate between S. stercoralis and third-stage hookworm larvae.

Line 214 – trophozoite? The methodology does not indicate that these evolutionary forms have been investigated.

Line 255 – Entamoeba hartmanni could be included because a simple morphological analysis cannot distinguish it from Entamoeba histolytica/dispar.

Line 257 – Table 2 Other studies in the same region found that Entamoeba coli was the second most prevalent species. What explanation would the authors give for its absence from this study? Did the Kato-Katz method make it possible to establish the prevalence of hookworm, while the two forms of culture made it possible to establish the prevalence of Strongyloides stercoralis only or also of hookworm?

Typing mistake - Mansoniella perstans - Mansonella perstans

Trichiurus trichiurusTrichuris trichiura

Line 351 - Bustinduy A.L. – Please, delete A.L.

Line 360 - combat polyparasitism and monoparasitism.

Author Response

Reviewer #1

Parasitosis can be understood as an indirect index of poor socio-economic development. Intestinal parasitic infections pose a significant public health risk, particularly among vulnerable groups such as children. Endoparasites in children are generally associated with challenges in child development (height, weight, and cognitive development). Addressing these issues is a major challenge in the context of One Health. The present paper was designed to estimate the prevalence and associated factors for polyparasitism and its effect on hemoglobin levels in school-aged children in Lambaréné and the sur- rounding villages in Gabon, Central Africa. Several publications on children in Gabon have already demonstrated high rates of polyparasitism in the analyzed population, including double or triple infections. Other data has also revealed a link between polyparasitism and anemia. Several studies aimed to determine the prevalence of co-infection with malaria and intestinal parasites and assess its association with anemia in school-aged children from rural and urban settlements in Gabon. To summarize, the issue raised by the authors is not unique to Gabon. For instance, the results of an epidemiological survey conducted in Gabonese provinces were recently published (see the following link: https://doi.org/10.1371/journal.pntd.0013161). In addition to the risk factors cited for the prevalence of parasitic diseases among schoolchildren, guardians or parents' level of education is also a relevant factor. Was this question included in the questionnaire?

The title accurately and keywords reflect the topic of the paper, which is clearly organized in several sections, including abstract, introduction, descriptions of methods, results with tables of data, discussion, conclusion and references. The manuscript is well-written and easy to understand. Regarding references, citations is partially adequate because recent study was not indicated https://doi.org/10.1371/journal.pntd.0013161).

The manuscript raised some questions.

Reply: We thank the reviewer for the comments.

Line 77 – of soil-transmitted helminth instead of sol soil-transmitted helminth

Reply: We thank the reviewer for this remark. We corrected the sentence (Line 79).

Line 92 – malaria and filariasis

Reply: We thank the reviewer for this remark. We replaced malaria with plasmodium infection in the sentence (L94).

Line 148- malaria?

Reply: We thank the reviewer for this remark. We added plasmodium infection in the sentence (L148).

Line 152- Direct examination of stool samples is a technique with low sensitivity. However, it can be useful for detecting trophozoites in diarrhoeal samples of intestinal protozoa, as well as motile Strongyloides stercoralis larvae.

Reply: We thank the reviewer for this remark.

Lines 167-179 - How many coproculture or Harada-Mori techniques were performed? 676?

Reply: We thank the reviewer for the question. We performed 656 coproculture and Harada-Mori.

Lines 187-189 – The information should be detailed if anyone wants to reproduce the techniques described here. However, the authors do not specify the criterion used to differentiate between S. stercoralis and third-stage hookworm larvae.

Reply: We thank the reviewer for the suggestion. This section has been rephrased accordingly. See below the new formulation (Section 2.8.3).

“In this technique, we used a filter paper to isolate also infective third-stage larvae. One gram of fresh faeces was placed at the centre of a sterile filter paper strip, which was then inserted into a 15 mL centrifuge tube containing 4 mL of distilled water. The tapered end of the strip touched the bottom of the tube, while the faecal spot remained above the water level to prevent direct immersion. The tube was sealed with a screw cap and incubated upright at 25–28 °C for 10 days. After incubation, the fluid at the bottom of the tube was centrifuged, and a smear was prepared from the resulting pellet. Microscopic examination was performed using a 10× objective lens to identify motile third-stage larvae. [24].To differentiate between Strongyloides stercoralis and hookworm larvae, in both culture techniques (Coproculture and Harada-Mori), we applied the following morphological criteria. Hookworm L3 larvae were identified by their pointed tail, short oesophagus, and overall smooth, tapered posterior end. S. stercoralis L3 larvae were distinguished by a notched tail, a long oesophagus with a 1:1 ratio to the intestine, and more vigorous motility [28].

Line 214 – trophozoite? The methodology does not indicate that these evolutionary forms have been investigated.

Reply: This applied only to Plasmodium falciparum infection. We have now revised the sentence to be clearer (section 2.8.6).

Line 255 – Entamoeba hartmanni could be included because a simple morphological analysis cannot distinguish it from Entamoeba histolytica/dispar.

Reply: We thank the reviewer for the remark the Entamoeba hartmanni has been added (Table 2).

Line 257 – Table 2 Other studies in the same region found that Entamoeba coli was the second most prevalent species.

What explanation would the authors give for its absence from this study?

Reply: Entamoeba coli was also found in our cohort, but it was not presented in the table because we focused only on pathogen protozoa. We have added it now to table 2.

 Did the Kato-Katz method make it possible to establish the prevalence of hookworm, while the two forms of culture made it possible to establish the prevalence of Strongyloides stercoralis only or also of hookworm?

Reply: The Hookworm prevalence was done with the Kato-Katz results, and the cultures used for Strongyloides stercoralis 

Typing mistake - Mansoniella perstans - Mansonella perstans

Trichiurus trichiurus – Trichuris trichiura

Reply: We thank the reviewer for the remark. This was corrected.

Line 351 - Bustinduy A.L. – Please, delete A.L.

Reply: We thank the reviewer for the suggestion. We removed it as suggested (Line 359).

Line 360 - combat polyparasitism and monoparasitism.

Reply: We thank the reviewer for the suggestion (Line 369)

Reviewer 2 Report

Comments and Suggestions for Authors

I reviewed with interest the manuscript by Nguema-Moure et al. on the infect of multiple and simultaneous parasitic infections on the level of haemoglobin among children in Gabon

This study is well written, but its presentation demands some adaptations in order to increase its value for the readers.

Title – adapt to read as: Polyparasitic INFECTIONS: Associated Factors and Effect on the Haemoglobin Level Among Children Living in Lambaréné Remote and Surrounding Rural Areas from Gabon – a cross-sectional study

  1. Why was the age range of 2 to 17 years selected? Please explain in the Materials and Methods section

  1. Participants: 656 or 676?

  1. How were the authors able to identify Plasmodium falciparum at the species level?

For presenting percentages, use one decimal place only – change accordingly throughout the manuscript

Keywords – display alphabetically

Line 43 – please give example(s) of some of these protozoa

Line 44 – people were not infected with schistosomiasis. They were infected with Schistosoma spp. or affected by schistosomiasis. Please correct

The same comment as above for the other diseases, especially for STH (line 45)

The expression “polyparasitic infections” is preferred to polyparasitism

Line 56 – Soil-Transmitted Helminths has already been abbreviated above as (STH) – please check line 39 – change accordingly throughout the manuscript for this and other abbreviations

Line 137 – why this exact number of 676?

Lines 138-140 – delete: ensuring that 138 the study results will be statistically significant and representative of the population under 139 study

Line 168 – hookworms (instead of Hookworms) – the same for line 188, etc.

Line 168 – Strongyloides stercoralis is also a STH – please mention this fact in text

Figure 2 – not clear: what is “physiological water”?

Lines 180-189 – use the past tense

Line 194 – write “eggs” in non-italic type

Line 204 – not clear: Malaria diagnosis using the blood thick smear [30].

Line 206 – replace “have been used” with “were used”

Line 215 – remove ellipsis (…)

Lines 215-217 – revise: respectively from STH species, IPS, urogenital schistosomiasis, Plasmodium spp. and blood filaria (Loa loa and/or Mansionella perstans) defined as the study parasites’ groups.

Lines 220-221 – revise: STH, IPS, Schistosoma spp., Plasmodium spp., or microfilariae worm species).

Statistical consideration or Statistical analysis?

Media and standard deviation were analysed – were data normally distributed? If not, medians should have been analysed

Write “regression” instead of Regression

Table 1 – add “years” below Characteristic

Abbreviate species names after their first use in the main text. Exceptions: beginning of sentences, and tables or figures

Lines 248-256 – this is not the best way to present confidence intervals – please revise by removing square brackets

Table 2 – please revise title to read as: Prevalence and parasite count means of soil-transmitted helminths (STH), Schistosoma haematobium, Plasmodium falciparum, and blood FILARIAL species in the study population.

Display parasite species by decreasing order of frequency within each section (i.e. STH, etc.)

How do the authors know that the species is Plasmodium falciparum?

Line 263 – define OR and b

Lines 271-277 – use the past tense

Table 3 – data about Age seem to be incomplete

Correct polyparasitism

Table 4 – data about Age seem to be incomplete

Discussion – abbreviate species names; use abbreviation STH

Line 351 – correct citation as: Bustinduy et al. [43]

References are not properly presented and should be standardized

Comments on the Quality of English Language

 The English could be improved to more clearly express the research.

Author Response

Reviewer #2

I reviewed with interest the manuscript by Nguema-Moure et al. on the infect of multiple and simultaneous parasitic infections on the level of haemoglobin among children in Gabon

This study is well written, but its presentation demands some adaptations in order to increase its value for the readers.

Title – adapt to read as: Polyparasitic INFECTIONS: Associated Factors and Effect on the Haemoglobin Level Among Children Living in Lambaréné Remote and Surrounding Rural Areas from Gabon – a cross-sectional study.

 Reply: We thank the reviewer for the suggestion

  1. Why was the age range of 2 to 17 years selected? Please explain in the Materials and Methods section

 Reply: We thank the reviewer for the comment. The main study from which this sub-analysis is done enrolled school-aged children of 2 to 17 years. The material and method contain the following details

“The original data came from a clinical trial aiming to assess the efficacy of a new regimen for the treatment of soil-transmitted helminth in school-age children, aged 2 to 17 years, from Lambaréné (both semi-urban and surrounding rural areas in Moyen-Ogoouée province) and remote areas of Tsamba-Magotsi (Ngounié province) ”’

Participants: 656 or 676?

 Reply: We thank the reviewer for the comment. The typo has been corrected.

  1. How were the authors able to identify Plasmodium falciparum at the species level?

Reply: We thank the reviewer for the comment. The identification of P. falciparum was performed using Giemsa-stained (10%) thick and thin blood smears, which were examined under a microscope. We rephrased section 2.8.6 for more precision.

For presenting percentages, use one decimal place only – change accordingly throughout the manuscript.

 Reply: We thank the reviewer for the comment.

Keywords – display alphabetically

 Reply: We thank the reviewer for this suggestion. This was done as suggested.

Line 43 – please give example(s) of some of these protozoa

 Reply: We thank the reviewer for this suggestion. The following sentence has been added (Line 45-46).

“Protozoa like: Blastocystis hominis, Giardia duodenalis, among others.  [7-8] also contribute to polyparasitic infection

Line 44 – people were not infected with schistosomiasis. They were infected with Schistosoma spp. or affected by schistosomiasis. Please correct

 Reply: We thank the reviewer for this remark. We corrected it as suggested (L46).

The same comment as above for the other diseases, especially for STH (line 45)

 Reply: We thank the reviewer for this remark. We corrected it as suggested.

The expression “polyparasitic infections” is preferred to polyparasitism

Line 56 – Soil-Transmitted Helminths has already been abbreviated above as (STH) – please check line 39 – change accordingly throughout the manuscript for this and other abbreviations

 Reply: We thank the reviewer for this remark. We replaced polyparasitism with polyparasitic infections in the manuscript.

Line 137 – why this exact number of 676?

 Reply: The estimated sample size was 550; Nevertheless, we extended the recruitment until the planned period. At the end of the recruitment, a total of 700 were screened.  Finally, 656 were enrolled as presented in the participant flow chart. Thanks for the comment the number typo has been corrected

Lines 138-140 – delete: ensuring that 138 the study results will be statistically significant and representative of the population under 139 study

  Reply: We thank the reviewer for this remark.

Line 168 – hookworms (instead of Hookworms) – the same for line 188, etc.

  Reply: We thank the reviewer for this remark. We corrected it.

Line 168 – Strongyloides stercoralis is also a STH – please mention this fact in text.

  Reply: We thank the reviewer for this suggestion.  

Figure 2 – not clear: what is “physiological water”?

   Reply: We thank the reviewer for this remark. We replace it with the Normal saline solution (0.9% NaCl)

Lines 180-189 – use the past tense

 Reply: We thank the reviewer for this remark.

Line 194 – write “eggs” in non-italic type

 Reply: We thank the reviewer for this remark.

Line 204 – not clear: Malaria diagnosis using the blood thick smear [30].

 Reply: We thank the reviewer for this remark. We rephrased this section accordingly.

Line 206 – replace “have been used” with “were used”

 Reply: We thank the reviewer for this remark.

Line 215 – remove ellipsis (…)

 Reply: We thank the reviewer for this remark.

Lines 215-217 – revise: respectively from STH species, IPS, urogenital schistosomiasis, Plasmodium spp. and blood filaria (Loa loa and/or Mansionella perstans) defined as the study parasites’ groups.

 Reply: We thank the reviewer for this remark. We revised it as suggested

Lines 220-221 – revise: STH, IPS, Schistosoma spp., Plasmodium spp., or microfilariae worm species.

 Reply: We thank the reviewer for this suggestion.

Statistical consideration or Statistical analysis?

 Reply: We thank the reviewer for this remark. This has been corrected accordingly

Media and standard deviation were analysed – were data normally distributed? If not, medians should have been analysed

  Reply: We thank the reviewer for this remark. The normality of continuous variables was assessed using the Kolmogorov-Smirnov test, the Shapiro-Wilk test. The results have been corrected accordingly

Write “regression” instead of Regression

 Reply: We thank the reviewer for this remark. We revised it as suggested at Line 246.

Table 1 – add “years” below Characteristic

 Reply: We thank the reviewer for this remark. We revised it.

Abbreviate species names after their first use in the main text. Exceptions: beginning of sentences, and tables or figures

 Reply: We thank the reviewer for this remark.

Lines 248-256 – this is not the best way to present confidence intervals – please revise by removing square brackets.

Reply: We thank the reviewer for this remark. This has been corrected

Table 2 – please revise title to read as: Prevalence and parasite count means of soil-transmitted helminths (STH), Schistosoma haematobium, Plasmodium falciparum, and blood FILARIAL species in the study population.

Reply: We thank the reviewer for the suggestion. It has been considered

Display parasite species by decreasing order of frequency within each section (i.e. STH, etc.)

 Reply: We thank the reviewer for this remark. This suggestion was taken into consideration.

How do the authors know that the species is Plasmodium falciparum?

 Reply: We thank the reviewer for this question. A thin blood smear was done in addition to the thick blood smear. The morphology was used to identify species.

Line 263 – define OR and b

Reply: We thank the reviewer for this remark. OR: (odd ratio) and β (regression coefficient) were defined under Table 3.

Lines 271-277 – use the past tense

 Reply: We thank the reviewer for this remark.

Table 3 – data about Age seem to be incomplete

  Reply: Thank you. The ages are in “years” and we used the median and IQR( in brackets)

Correct polyparasitism

 Reply: We thank the reviewer for this remark.

Table 4 – data about Age seem to be incomplete

Reply: We corrected it now.Thank

Discussion – abbreviate species names; use the abbreviation STH

 Reply: We thank the reviewer for the suggestion, and the discussion has been corrected accordingly

Line 351 – correct citation as: Bustinduy et al. [43]

 Reply: We thank the reviewer for this remark. This was modified.

References are not properly presented and should be standardised

Reply: We thank the reviewer for this remark. This was modified.

Reviewer 3 Report

Comments and Suggestions for Authors

Author Response

  1. The Authors didn’t mention the procedures for infected and anaemic children. Did they provide the treatment or were referred to the local public health institution? This information needs to be included.

Reply : The following sentence has been added

“Blood samples were collected by venepuncture  in EDTA tubes (2.7 mL). Before microscopic examination, haematological parameters were analysed using the Yumizen H500 automated system. Haematological printing results are sent within 24 hours to the clinician for assessment and decision.”

  1. Line 45 – STH must be cited first in full and then abbreviated by the acronym.

Reply : Thank you. STH has been cited first in the first paragraph

  1. Line 88 – Blood samples were collected by venepuncture, wright?

Reply: Yes. We added this information now. Thanks

  1. Line 169 – The description of coproculture technique in the text is slightly different from the Fig.2 - stool spread in the filter paper is the correct procedure.

Reply : Thank you. It has been corrected  

  1. Line 194 – eggs not in italic

Reply : Thank you. It has been corrected  

  1. Line 216 – I suggest to use S. haematobium instead “urogenital schistosomiasis”, as all others pathogenic agents are cited by group species.

Reply: The paragraph has been corrected. Thank you

  1. Throughout the text, parasite’s genera and species names should be in italics (ex. Schistosoma, Plasmodium, Ascaris lumbricoides - Lines: 216, 220, 305)

Reply: Thank you. It has been corrected

  1. Lines 253, 312 – Giardia duodenalis or G. intestinalis? Please, uniform.

Reply: Thank you. It has been corrected

  1. Line 323 – rural oguoouée? See previous sentence.

Thank you. The typo has been corrected

  1. Please, uniform the references according to journal guidelines. Publications are cited differently, some are in full and others in the abbreviate form (i.e. #488; #506)

Reply : The reference list has been revised

  1. Line 454 – No Authors?

Reply : The reference list has been revised

Round 2

Reviewer 2 Report

Comments and Suggestions for Authors

The authors have addressed my comments and accepted my suggestions.